# Comprehensive Analysis Reveals That ISCA1 Is Correlated with Ferroptosis-Related Genes Across Cancers and Is a Biomarker in Thyroid Carcinoma

**DOI:** 10.3390/genes15121538

**Published:** 2024-11-28

**Authors:** Dejun Xiong, Zhao Li, Ling Zuo, Juan Ge, Yuhan Gu, Erhao Zhang, Xiaorong Zhou, Guiping Yu, Mengmeng Sang

**Affiliations:** 1Department of Immunology, School of Medicine, Nantong University, Nantong 226019, China; 2331310020@stmail.ntu.edu.cn (D.X.); lz15394673379@126.com (Z.L.); zling_ntu@163.com (L.Z.); 2231310014@stmail.ntu.edu.cn (J.G.); 2323110038@stmail.ntu.edu.cn (Y.G.); zhangerhao@ntu.edu.cn (E.Z.); zhouxiaorong@ntu.edu.cn (X.Z.); 2Department of Respiratory Medicine, Affiliated Nantong Hospital of Shanghai University, Nantong 226011, China; 3Department of Cardiothoracic Surgery, Jiangyin People’s Hospital Affiliated to Nantong University, Jiangyin 214499, China

**Keywords:** ISCA1, pan-cancer, ferroptosis-related genes, drug prediction, thyroid carcinoma

## Abstract

Background: *ISCA1* (Iron–Sulfur Cluster Assembly 1) is involved in the assembly of iron–sulfur (Fe–S) clusters, which are vital for electron transport and enzyme activity. Some studies suggest the potential involvement of *ISCA1* in tumor progression through interactions with ferroptosis-related genes (FRGs) and the tumor immune microenvironment (TME). However, there has been no systematic analysis of its role in FRGs and the TME or its predictive value for prognosis and immunotherapy response across different cancer types. Methods: In this study, we analyzed the expression and prognosis of *ISCA1* RNA, CNV, methylation, and protein in multiple tumor tissues via data from the TCGA and CPTAC databases and clinical information. We conducted a comprehensive analysis of the correlations between *ISCA1* and FRGs, immune-related genes (including immune regulatory genes and immune checkpoint genes), immune cell infiltration, immune infiltration scores, tumor stemness, and genomic heterogeneity. Results: We performed drug prediction and validation through molecular docking and molecular dynamics analysis to identify candidate drugs that could promote or inhibit *ISCA1* RNA expression. Our findings revealed that *ISCA1* could serve as a biomarker in thyroid carcinoma, play a role with different FRGs in various cell types, and mediate different ligand–receptor pathways for cell–cell communication. Conclusions: Overall, our study highlights the potential of *ISCA1* as a novel biomarker for predicting prognosis and immunotherapeutic efficacy in thyroid carcinoma and suggests its potential for developing novel antitumor drugs or improving immunotherapy.

## 1. Introduction

Cancer is becoming a major public health issue worldwide, with standardized incidence rates for all types of cancer in China increasing by an average of 1.4% per year [1]. Ferroptosis is a type of programmed cell death characterized by the iron-dependent accumulation of lipid peroxides to lethal levels [2]. Many cancer cells are particularly susceptible to ferroptosis-inducing agents [3]. This susceptibility may be leveraged for cancer treatment, especially in cases where cancer cells are resistant to traditional therapies such as chemotherapy and radiation [4].

Cancer cells often exhibit alterations in iron metabolism characterized by increased iron uptake and storage [5]. Iron is essential for the formation of reactive oxygen species (ROS) through Fenton reactions, which directly contribute to the lipid peroxidation that drives ferroptosis [6]. The oxidation of polyunsaturated fatty acids (PUFAs) within phospholipids is a hallmark of ferroptosis [7]. Cancer cells frequently exhibit altered lipid metabolism with increased incorporation of polyunsaturated fatty acids (PUFAs), which are substrates for peroxidation, into their membranes [8]. Additionally, cancer cells often experience increased oxidative stress and are heavily reliant on the antioxidant system, which includes glutathione (GSH) and glutathione peroxidase 4 (GPX4) [8]. GPX4 uses GSH to neutralize lipid peroxides, and its inhibition is a critical trigger of ferroptosis [9]. Ferroptosis represents a novel and potentially powerful therapeutic avenue in cancer treatment. Research has shown that the induction of ferroptosis can overcome resistance to conventional therapies [8]. Furthermore, combining ferroptosis inducers with traditional therapies such as chemotherapy, radiation, or immunotherapy could increase the overall treatment efficacy [10].

Iron–sulfur cluster assembly 1 (*ISCA1*) is involved in the assembly of iron–sulfur (Fe–S) clusters, which are vital for various cellular functions, including electron transport and enzyme activity [11]. Fe–S clusters play a significant role in mitochondrial function and cellular energy metabolism [12]. Alterations in *ISCA1* could theoretically impact the availability of iron for processes necessary for ferroptosis, thus affecting cancer cell vulnerability to this cell death mechanism [13]. Given that mitochondria are central to oxidative stress regulation and that *ISCA1* impacts mitochondrial function through Fe–S cluster assembly, any dysfunction or alteration in *ISCA1* expression or activity might influence the mitochondrial production of ROS [11,14]. Elevated ROS levels can induce lipid peroxidation, leading to ferroptosis. In cancers, aberrant iron metabolism and mitochondrial dysfunctions are prevalent and have been implicated in both tumorigenesis and resistance to treatment [6]. If *ISCA1* is dysregulated, it could potentiate the dysregulation of iron and oxidative homeostasis, thus increasing or decreasing the susceptibility of cancer cells to ferroptosis, depending on the nature of the disruption. Should *ISCA1* influence ferroptosis susceptibility through its regulatory effects on iron metabolism and oxidative stress, it represents a potential therapeutic target. Modulating *ISCA1* could thus enhance the effectiveness of ferroptosis-inducing treatments, particularly in malignancies that are resistant to conventional therapies.

In this study, we employed public databases to examine the RNA expression, copy number variation (CNV), and methylation levels of *ISCA1* across various cancer types. Additionally, we elucidated the prognostic significance of *ISCA1* in terms of tumor outcomes. Moreover, we delineated the potential functions of *ISCA1* and its impact on ferroptosis-related genes (FRGs), immune regulatory genes (IRGs), immune checkpoint genes (ICGs), immune cell infiltration, tumor stemness, and genomic heterogeneity at the pan-cancer level. We also performed drug prediction, molecular docking, and molecular dynamics analyses to search for candidate compounds that could promote or inhibit *ISCA1* RNA expression. Our systematic analysis suggested that *ISCA1* could serve as a predictive biomarker for prognosis and response to tumor immunotherapy in thyroid carcinoma (THCA) patients. Moreover, we conducted an analysis of the regulatory mechanisms of *ISCA1* and FRGs in THCA, utilizing data from The Cancer Genome Atlas (TCGA) and single-cell databases. Our study also elucidated the regulatory roles of *ISCA1* in cell–cell communication and identified several significant ligand–receptor interactions. Collectively, our findings provide a comprehensive perspective on the regulatory functions of *ISCA1* and FRGs across various cancer types, potentially offering new insights for the development of novel therapeutic strategies.

## 2. Materials and Methods

### 2.1. Data Collection and Preprocessing

The bulk RNA expression data and associated clinical metadata were obtained from the Genotype-Tissue Expression (GTEx) database (https://www.gtexportal.org/home/, accessed on 28 June 2024) [15] and The Cancer Genome Atlas (TCGA, https://www.cancer.gov/ccg/research/genome-sequencing/tcga, accessed on 28 June 2024) databases [16]. The RNA data were provided in Transcripts Per Million (TPM) format, and the combat algorithm from the sva package (version 3.5) was employed to eliminate batch effects of RNA data from the GTEx and TCGA databases [17]. Protein expression data for various cancer types were retrieved from the Clinical Proteomic Tumor Analysis Consortium (CPTAC) database (https://pdc.cancer.gov/pdc/, accessed on 28 June 2024) [18]. The Wilcoxon rank sum test and signed rank test were used to analyze the differential expression of *ISCA1* between different groups.

### 2.2. Gene Mutation Analysis

The cBioPortal database (http://www.cbioportal.org/, accessed on 28 June 2024) was used to analyze the genetic and epigenetic alterations of *ISCA1* in multiple cancer types in the TCGA database [19]. Additionally, the copy number variants (CNVs) and methylation statuses of *ISCA1* within the TCGA dataset were retrieved from the Genomic Data Commons (GDC) database (https://portal.gdc.cancer.gov/, accessed on 28 June 2024) [20].

### 2.3. Cox Regression Analysis

The survival package (version 3.2-7) was used to evaluate the prognostic significance of *ISCA1* across various cancer types. This analysis incorporated multiple factors, including survival time, survival status, and *ISCA1* expression levels, which encompassed RNA expression, CNVs, and methylation levels [21].

### 2.4. Ferroptosis-Related and Immune-Related Gene Analysis

We compiled ferroptosis-related genes (FRGs) from the FreeDb website (V2, http://www.zhounan.org/ferrdb/current/, accessed on 28 June 2024) [22], including 369 driver genes, 348 suppressor genes, 11 marker genes, and 116 unclassified regulator genes. Additionally, we gathered immune regulatory genes from the study conducted by Hu et al. [23], which included 41 chemokine genes, 18 receptor genes, 21 major histocompatibility complex (MHC) genes, 24 immunoinhibitor genes, and 46 immunostimulator genes. The immune checkpoint genes, consisting of 24 inhibitory genes and 36 stimulatory genes, were also retrieved from the study by Thorsson et al. [24]. We obtained the RNA expression, CNV, and methylation values of these ferroptosis-related and immune-related genes from the GDC database.

### 2.5. Immune Infiltration, Tumor Stemness, and Genomic Heterogeneity Analysis

The Timer [25], EPIC [26], MCPCounter [27], QUANTISEQ [28], CIBERSORT [29], and xCell [30] algorithms from the IOBR (v 0.99.9) package [31] were used to evaluate the infiltration of 6, 8, 10, 11, 22, and 64 immune cells, respectively, for each patient of each cancer type on the basis of gene expression. Additionally, the ESTIMATE package (v 2.0.13) was used to calculate the stromal, immune, and ESTIMATE scores for each tumor patient on the basis of RNA expression [32].

The scores for DNA (DNA methylation-based), EREG-METH (epigenetically regulated DNA methylation-based), DMP (differentially methylated probe-based), ENH (enhancer element/DNA methylation-based), RNA (RNA expression-based), and EREG were obtained. EXPs (epigenetically regulated RNA expression-based) were calculated for each cancer patient by Malta et al.‘s research on the basis of RNA and methylation expression levels [33].

We downloaded the simple nucleotide variation data processed by *MuTect2* software [34] from the GDC database (https://portal.gdc.cancer.gov/, accessed on 28 June 2024) and calculated the TMB score for each cancer patient via the TMB function of the Maftools package (version 2.18) [35]. The InferHeterogeneity function of the Maftools package was used to calculate the Mutant-Allele Tumor Heterogeneity (MATH) score for each patient [36]. MSI (microsatellite instability) scores for each cancer patient were obtained from Bonneville et al.’s study. Furthermore, NEO (Neoantigen) scores, purity scores, ploidy scores, HRD (homologous recombination deficiency) scores, and LOH (loss of heterozygosity) scores for each cancer patient were gathered from Thorsson et al.’s study [24]. We then analyzed the correlations among RNA expression, CNV, *ISCA1* methylation levels, immune cell infiltration, immune infiltration scores, tumor stemness, and genomic heterogeneity.

### 2.6. Drug Prediction and Validation

We obtained the z-scale IC50 values for each chemical compound, along with RNA-seq data in log2 (FPKM+1) format for 60 cell lines, from the CellMiner database (https://discover.nci.nih.gov/cellminer/, accessed on 28 June 2024) [37]. We then estimated the correlations between *ISCA1* RNA expression and the activity z score data of each chemical compound. Chemical compounds with a *p*-value less than 0.01 were identified as potential drugs that either promoted or inhibited *ISCA1* RNA expression.

The chemical structures of the predicted drugs were downloaded from the ZINC15 database (https://zinc15.docking.org/, accessed on 28 June 2024) [38], whereas the protein structure of ISCA1 was obtained from the Protein Data Bank database (https://www.rcsb.org/, accessed on 28 June 2024) [39]. Both the chemical and protein structures were pretreated with UCSF Chimera (v 1.15) [40]. DOCK software (v 6.10) was used to predict the binding patterns of the small molecules and protein complexes [41].

We validated the interaction between proteins and active compounds via molecular docking analysis followed by molecular dynamics (MD) simulation. The Gromacs (v 2023) software was used for the simulation, with the physical conditions set at a constant temperature of 310 K and pressure of 101 kPa. Periodic boundary conditions were applied, and the TIP3P water model was used to simulate a 0.145 mol/L neutral sodium chloride solution in a human environment [42]. A 100 ns MD simulation was conducted via the Gromacs embedded program, with confirmation storage calculations performed every 10 ps. The results of the molecular dynamics simulation, including the Rg value (radius of gyration), RMSD (root mean square deviation), RMSF (root mean square fluctuation), and SASA (solvent-accessible surface area), were analyzed and visualized via the Gauss embedded program and DuIvyTools (v 0.4.0) software. The Rg parameter serves as a fundamental indicator of the total size of the chain molecule, providing insights into the variation in protein structure during molecular dynamics simulation [43]. The RMSD parameter was crucial for examining the equilibration and structural stability of the protease in the presence of a docked ligand by indicating the rate of the mean distance between atoms. Comparing the RMSD values of the protein and compound helped evaluate changes in protein molecular dynamics and the conformational stability of the protein–ligand complex [44]. The RMSF parameter enabled us to assess the flexibility of the protease in conjunction with a compound and identify local variations in the protein chain [45]. Finally, the SASA parameter plays a critical role in protein folding and stability research by illustrating changes in the folding and unfolding of complexes over time on the basis of the van der Waals contact surface of the molecule and a hypothetical center of a solvent sphere [46].

### 2.7. Differential Gene Expression (DEG) and GSVA Analysis

The limma-voom algorithm was utilized to analyze differentially expressed FRGs among subgroups of patients via bulk RNA-seq data [47]. Genes with an adjusted *p*-value of <0.05 and log2-fold change (log2FoldChange) greater than 0.584 or less than −0.584 were identified as DEGs. Additionally, GSVA was employed to assess various hallmark pathways between subgroups in unsupervised and parameter-free conditions [48]. The hallmark gene sets were derived from the Molecular Signatures Database (https://www.gsea-msigdb.org/gsea/msigdb/index.jsp, accessed on 28 June 2024).

### 2.8. Anticancer Immune Response Analysis

The Tracking Tumor Immunophenotype (TIP) database (http://biocc.hrbmu.edu.cn/TIP/, accessed on 28 June 2024) was used to analyze the anticancer immune response between subgroups [49]. Additionally, we calculated the correlations between the scores of the cancer-immunity cycles and *ISCA1* RNA expression in different groups.

### 2.9. Single-Cell Data Analysis

The GSE191288 data concerning thyroid carcinoma (THCA) were used for further analysis [50]. The Seurat (v 4.4.0) package was used to analyze the scRNA-seq data [51]. The FindMarker function was applied to discern DEGs and differentially expressed FRGs between subgroups with a *p*-value less than 0.05 and a log2-fold change (log2FoldChange) greater than 0.584 or less than −0.584. We classified activated and nonactivated cells on the basis of whether *ISCA1* is expressed in each cell type. The CellChat (v 1.6.1) package was subsequently employed to simulate cellular communication by combining the interaction between receptors and their cofactors [52]. The AUCell package (v 1.22.0) was used to calculate the scores for ferroptosis driver genes, ferroptosis suppressor genes, ferroptosis marker genes, ferroptosis unclassified regulator genes, and FRGs [53]. The scores were compared between subgroups in each cell type.

### 2.10. Statistical

We utilized R (v 4.2.3) software for data processing, statistical analysis, and plotting. The specific statistical methods used are detailed in the abovementioned methods.

## 3. Results

### 3.1. The Expression Levels of ISCA1 in Normal and Tumor Tissues

The GTEx database was used to investigate the RNA expression levels of *ISCA1* in normal human tissues. The results indicated that *ISCA1* was relatively greater in brain organs, including the cerebellar hemisphere, frontal cortex, and cerebellum, but lower in whole blood and the pancreas (Appendix A). Furthermore, we explored the results of scRNA-seq data from several normal tissues and showed that *ISCA1* was significantly expressed in stromal cells, including endothelial cells and fibroblasts (Appendix A). *ISCA1* is located on chromosome 9 and generates three distinct transcripts on the basis of GRCh38 (V46). We examined the differential expression of the *ISCA1* gene and transcripts in normal and cancer tissues (Figure 1A–D). We found that *ISCA1* was significantly more highly expressed in the tumor tissues of CHOL, ESCA, GBM, HNSC, LAML, LGG, LIHC, LUSC, PAAD, PRAD, SKCM, and STAD, whereas it was more highly expressed in the normal tissues of BLCA, CESC, KICH, KIRC, KIRP, OV, READ, TGCT, THCA, UCEC, and UCS (Figure 1A). Compared with those of the other transcripts, the expression levels of ENST00000375991 were notably greater in tissues. The three transcripts were significantly different between normal and tumor tissues in THCA and HNSC (*p* < 0.01), but there was no trend in PAAD, PCPG, PRAD, SARC, or SKCM (Figure 1B–D).

We subsequently comprehensively evaluated the expression of *ISCA1* in the clinicopathological features of the tumors (Figure 1E–K). *ISCA1* and three transcripts were more highly expressed in the young groups of LUSC (Figure 1E). *ISCA1*, ENST00000375991, and ENST00000326094 were significantly differentially expressed between the T periods of PRAD (Figure 1F). *ISCA1* was highly expressed in the M0 period of KIRC and PRAD, and ENST00000311534 was highly expressed in M0 of TGCT but expressed at low levels in M0 of ACC (Figure 1G). *ISCA1* and three transcripts were significantly differentially expressed between various N periods in PRAD and THCA (Figure 1H). *ISCA1* was differentially expressed between various grades of CESC, HNSC, and KIRC, whereas ENST00000311534 was differentially expressed in UCEC (Figure 1I). Significant differences in the expression of *ISCA1* and three transcripts were detected between various stages of THCA (Figure 1J). Furthermore, the expression of *ISCA1* differed between males and females in KIRC, whereas ENST00000375991 was differentially expressed in ACC, and ENST00000326094 was differentially expressed in BRCA (Figure 1K).

We used the UALCAN database to analyze *ISCA1* protein levels between normal and tumor tissues. The *ISCA1* protein was significantly upregulated in normal tissues of BRCA, GBM, LIHC, and PAAD patients, whereas it was downregulated in normal tissues of COAD, HNSC, LUAD, LUSC, OV, and UCEC patients (Appendix A). We conducted a detailed analysis of the genomic aberrations and CNVs of *ISCA1* across various cancer types. Our study revealed that the frequency of alterations in *ISCA1* was highest in SARC, exceeding 3% (Appendix A). Mutations in *ISCA1* occurred only in COAD, LAML, LUAD, LUSC, SARC, SKCM, and UCEC, with relatively low mutation frequencies (Appendix A). When we analyzed the correlation between *ISCA1* CNV, methylation and RNA levels, we found that the CNV of *ISCA1* was positively correlated with RNA expression in most cancer types (Appendix A), whereas the methylation levels of *ISCA1* were negatively related to RNA expression in most cancer types (Appendix A).

### 3.2. The Prognostic Impact of ISCA1 Across Cancers

To investigate the prognostic role of *ISCA1* expression in different human cancers, survival data, including overall survival (OS), disease-specific survival (DSS), disease-free interval (DFI), and progression-free interval (PFI), were analyzed. The expression levels of *ISCA1* RNA were positively correlated with OS in BLCA, BRCA, CESC, GBM, KIRC, LAML, LGG, OV, READ, THCA, and THYM patients in terms of age, and positively correlated with OS in KIRC, LGG, and PAAD patients in terms of grade. Furthermore, *ISCA1* RNA expression was correlated with OS in BLCA, BRCA, COAD, ESCA, HNSC, KICH, KIRC, KIRP, LUAD, and STAD patients in relation to stage and the M, N, and T periods (Figure 2A, Appendix A). Additionally, the expression level of *ISCA1* RNA was found to be a protective factor in KIRC but a risk factor in ACC, HNSC, THCA, and UCS on the basis of OS. It was also a protective factor in KIRC patients but was a risk factor for ACC and THCA patients in terms of DSS. In addition, it was identified as a risk factor in LUSC patients on the basis of DFS and a protective factor in KIRC, PAAD, and UCEC patients, but a risk factor in ACC, HNSC, PRAD, and UVM patients on the basis of the DFI (Figure 2B, Appendix A). The *ISCA1* CNV was identified as a risk factor in terms of OS, DSS, and PFI in the ACC but acted as a protective factor in terms of OS, DSS, DFI, and PFI in UCEC (Figure 2B, Appendix A). Furthermore, *ISCA1* methylation was found to be a risk factor for OS and DSS in CESC and LUAD patients (Figure 2B, Appendix A). The correlations between the RNA expression of the three *ISCA1* transcripts and survival were subsequently analyzed. The results indicated that ENST00000311534 was a risk factor for OS in ACC and HNSC patients, DSS in ACC patients and CESC patients, DFS in LUSC patients, and PFI in ACC patients (Figure 2C, Appendix A). ENST00000326094 was identified as a risk factor for DSS in the ACC, and PFI in the ACC and CESC (Figure 2C, Appendix A). Furthermore, ENST00000375991 was found to be a risk factor in terms of DFS in HNSC and LUSC patients, as well as in the PFI in ACC and HNSC patients (Figure 2C, Appendix A).

### 3.3. The Correlations Between ISCA1 and Ferroptosis-Related Genes (FRGs), Immune Regulatory Genes (IRGs), and Immune Checkpoint Genes (ICGs)

To investigate the potential relationships between *ISCA1* and ferroptosis, we first calculated scores for ferroptosis-driver genes, ferroptosis-marker genes, ferroptosis-suppressor genes, and ferroptosis-unclassified genes via single-sample gene set enrichment analysis (ssGSEA) algorithms with RNA expression data. These scores were then defined as driver, marker, suppressor, and unclassified scores. We estimated the correlations between *ISCA1* RNA expression and the four ferroptosis scores and found that *ISCA1* was negatively correlated with at least three scores in PAAD and THCA patients but positively correlated with at least three scores in LAML, OV, THYM, and UVM patients (*p* < 0.05) (Figure 3A). *ISCA1* CNV was negatively correlated with at least three scores in KIRC and LIHC (Appendix A). *ISCA1* methylation was negatively correlated with at least three scores in BLCA, BRCA, KIRP, and STAD patients but positively correlated with at least three scores in GBM, LIHC, and TGCT patients (Appendix A). ENST00000311534 was negatively correlated with at least three scores in TGCTs but positively correlated with at least three scores in BRCA, CHOL, KICH, KIRP, LAML, OV, PCPG, PRAD, and STAD (*p* < 0.05) (Figure 3B). Similarly, ENST00000375991 was positively correlated with at least three scores in BLCA, COAD, HNSC, KIRP, LAML, LUAD, OV, PRAD, THYM, and UCES (*p* < 0.05) (Figure 3C). Additionally, ENST00000326094 was negatively correlated with at least three scores in STAD but positively correlated with at least three scores in HNSC, LIHC, OV, PRAD, THYM, and LUSC (*p* < 0.05) (Figure 3D).

We subsequently calculated the correlations between the RNA expression of *ISCA1* and FRGs (Appendix A). Surprisingly, we identified 49 FRGs, including 29 drivers, 15 suppressors, two markers, and three unclassified regulator genes that were significantly correlated with *ISCA1* in more than 28 cancer types (33 in total), with *p* values less than 0.05 (Figure 3E). Among the 49 FRGs, *HSPB1* was the only gene that was negatively correlated with *ISCA1* in most cancer types (Figure 3A). We subsequently investigated the correlations between the CNV of *ISCA1* and the RNA expression of FRGs and reported that the *ISCA1* CNV was positively correlated with *TSC1*, *FXN*, *BRD3*, and *MAPKAP1* (Appendix A) in most cancer types. We also calculated the relationships between the methylation levels of *ISCA1* and FRG RNA expression in multiple cancer types, and the results revealed that *ISCA1* methylation was negatively correlated with most of these 49 FRGs in BLCA and PAAD (Appendix A). Furthermore, we analyzed the relationships between three *ISCA1* transcript isoforms and FRG RNA expression and found positive correlations between the most predominant transcript isoform, ENST00000375991, and FRGs, which were stronger than those of the other two transcripts (Figure 3F–H, Appendix A).

We further investigated the regulatory relationships between *ISCA1* and immune genes, including immune regulatory genes (IRGs) and immune checkpoint genes (ICGs). The ssGSEA algorithm was used to calculate the gene set scores for inhibitory, stimulatory, chemokine, immunoinhibitor, immunostimulator, MHC, and receptor genes. *ISCA1* RNA was found to be negatively correlated with at least five immune gene set scores in the ACC, BRCA, CESC, LGG, SARC, SKCM, and THCA but positively correlated with at least five immune gene set scores in the UVM (*p* < 0.05) (Figure 4A). *ISCA1* CNV was negatively correlated with at least five immune gene set scores in the ACC but positively correlated with those in the BLCA, CESC, HNSC, and SARC (Appendix A). *ISCA1* methylation was negatively correlated with at least five immune gene set scores in BLCA and PAAD but positively correlated in COAD, DLBC, GBM, HNSC, KIRC, LIHC, LUAD, LUSC, MESO, SKCM, and TGCT (Appendix A). ENST00000311534 was negatively correlated with at least five immune gene set scores in LGG, LUSC, SKCM, and THCA but positively correlated with STAD (*p* < 0.05) (Figure 4B). Similarly, ENST00000375991 was negatively correlated with at least five immune gene set scores in ACC, BRCA, CESC, KIRC, KIRP, LGG, SARC, SKCM, and THCA but positively correlated in BLCA, COAD, and PRAD (*p* < 0.05) (Figure 4C). Additionally, ENST00000326094 was negatively correlated with at least five immune gene set scores in the ACC, BRCA, CESC, GBM, KIRP, LGG, LIHC, LUSC, SARC, SKCM, STAD, and THCA but positively correlated in the DLBC (*p* < 0.05) (Figure 4D). The RNA expression levels of *ISCA1* and three transcripts were significantly correlated with all seven immune gene set scores in THCA patients (*p* < 0.0001).

We then analyzed the correlations of RNA expression between *ISCA1* and immune genes (Appendix A). A total of 33 immune genes, including six inhibitory genes, eight stimulatory genes, two chemokines, six immunoinhibitors, four immunostimulators, five MHC, and two receptor genes, were correlated with *ISCA1* in more than 15 cancer types (*p* < 0.05) (Figure 4E). *ISCA1* was correlated with *ENTPD1*, *HMGB1*, *TLR4*, and *TGFBR1* in more cancers (Figure 4E). Compared with other cancers, *ISCA1* was more closely related to these 33 immune genes in UVM (Figure 4E). We then further analyzed the correlations between CNV, methylation of *ISCA1*, and RNA expression of immune genes (Appendix A). The results revealed that *ISCA1* CNV was positively correlated with *TGFBR1* RNA expression in most cancers (Appendix A), and *ISCA1* methylation was correlated with the RNA expression levels of these 33 immune genes in COAD, LIHC, and TGCT (Appendix A). Furthermore, we estimated the relationships of RNA expression between three transcripts and immune genes (Appendix A). ENST00000311534 was significantly correlated with *HMGB1* (Figure 4F), ENST00000326094 was significantly correlated with *BTN3A1*, *CD160*, *TGFBR1*, and *IL6R* (Figure 4G), and ENST00000375991 was significantly correlated with *CD276*, *ENTPD1*, *HMGB1*, *TLR4*, *KDR*, and *TGFBR1* among these 33 immune genes in more than 20 cancer types (Figure 4H). Moreover, ENST00000311534 was significantly correlated with more than 20 of these 33 immune genes in BRCA, KIRC, LGG, PCPG, and STAD (Figure 4F); ENST00000326094 was significantly correlated with DLBC, KIRC, OV, PRAD, and THYM (Figure 4G); and ENST00000375991 was significantly correlated with BLCA, DLBC, HNSC, KIRC, KIRP, LGG, PAAD, PCPG, PRAD, THCA, and THYM (Figure 4H).

### 3.4. Effect of ISCA1 Expression on the Immune Microenvironment

To investigate the influence of *ISCA1* expression on the immune microenvironment, various algorithms, including CIBERSORT, EPIC, MCPcounter, QUANTISEQ, TIMER, and xCell, were utilized to estimate immune cell infiltration in human cancers, and Spearman correlation analyses were subsequently performed (Appendix A). The correlations between *ISCA1* RNA expression and immune cell infiltration were calculated and are displayed in Appendix A. The heatmap in Figure 5A illustrates that immune cells were significantly correlated with *ISCA1* RNA expression, with a *p*-value less than 0.05 in more than 20 cancer types. Furthermore, Tregs (regulatory T cells) from CIBERSORT, CAFs (cancer-associated fibroblasts) from EPIC, and NK cells from QUANTISEQ, as well as CD4+ Tem cells, epithelial cells, mesenchymal stem cells (MSCs), mv endothelial cells, neurons, natural killer T cells (NKTs), plasma cells, and smooth muscle cells from the xCell algorithm, were identified. Overall, the RNA expression of *ISCA1* was positively related to the infiltration of NK cells, neurons, and smooth muscle cells but negatively correlated with that of other cells in most cancer types (Figure 5A, Appendix A). The infiltration of these immune cells had a relatively minor effect on the CNV of *ISCA1*, as depicted in Appendix A. In addition, infiltration of more than eight of these immune cells was significantly associated with the methylation level of *ISCA1* in HNSC and LUAD (Appendix A). Moreover, the infiltration of more than eight of these immune cells was correlated with the RNA expression of ENST00000311534 in KIRC, KIRP, SKCM, and THCA (Figure 5B); the RNA expression of ENST00000375991 in BLCA, BRCA, HNSC, KIRC, KIRP, LIHC, LUAD, LUSC, PAAD, PRAD, SKCM, STAD, THCA, and UCEC (Figure 5C); and the RNA expression of ENST00000326094 in BRCA, KIRC, LGG, PRAD, SKCM, and THCA (Figure 5D). Furthermore, CAFs from the EPIC algorithm, CD4 T cells from the TIMER algorithm, and Th1 cells from the xCell algorithm were found to correlate with the RNA expression of ENST00000326094 and ENST00000375991 in more than 20 cancer types (Appendix A).

The *ESTIMATE* package was utilized to estimate the stromal, immune, and ESTIMATE scores for each patient. We analyzed the correlations between *ISCA1* expression and the three immune infiltration scores and found that the three immune infiltration scores were negatively correlated with *ISCA1* RNA expression in THCA, SKCM, LIHC, LGG, and UCEC (Figure 5E). Additionally, the three immune infiltration scores positively impacted CNVs of *ISCA1* in BLCA and SRAC but had a negative effect in ACC and KIRC (Appendix A). Furthermore, these immune infiltration scores were positively correlated with *ISCA1* methylation levels in COAD, ESCA, HNSC, KIRC, LIHC, LUAD, LUSC, MESO, SKCM, and TGCT but negatively correlated in BLCA and PAAD (Appendix A). We then focused on the correlations between the RNA expression of the three isoforms and the three immune infiltration scores. ENST00000311534 was negatively related to ESCA, LGG, LUSC, SKCM, and THCA (Figure 5F). ENST00000326094 was negatively related to ACC, GBM, KIRP, LGG, LIHC, LUSC, SKCM, THCA, and UCEC (Figure 5G). ENST00000375991 was negatively related to ACC, LGG, SKCM, THCA, and UCEC (Figure 5H). The RNA expression of *ISCA1* and its three transcripts was negatively correlated with the three immune infiltration scores in LGG, SKCM, and THCA. These results suggest that *ISCA1* RNA expression plays a significant role in immune regulation in these three types of cancer.

### 3.5. ISCA1 Influences Tumor Stemness and Genomic Heterogeneity

We analyzed the correlations between the RNA expression, CNV, and methylation levels of *ISCA1* and tumor stemness in each cancer type. We found that *ISCA1* RNA was positively correlated with all six tumor stemness features in TGCTs and was positively correlated with DMPs, DNAss, ENHss, and EREG.METH in PCPGs but was negatively correlated with BLCA and BRCA (Figure 6A, Appendix A). The *ISCA1* CNV negatively affected more than four tumor stemness features in BLCA, BRCA, ESCA, HNSC, KIRP, LUAD, LUSC, and UVM but positively affected them in SARC (Appendix A). ISCA1 methylation was negatively related to six tumor stemness features in COAD, DLBC, HNSC, KIRP, LIHC, LUAD, LUSC, and TGCT and positively related to DMPs, DNAss, ENHss, and EREG.METH was negatively related to BLCA, LGG, and PAAD but negatively related to CESC, ESCA, KIRC, LIHC, PCPG, SARC, and SKCM (Appendix A). We also further analyzed the correlations between the RNA expression of these three transcripts and tumor stemness. ENST00000311534 negatively influenced more than four features in KIRP, LGG, and OV but positively influenced LAML, LUAD, SARC, TGCT, and THYM (Figure 6B, Appendix A). ENST00000311534 was positively correlated with EREG.EXPs and RNAs in more than 15 cancer types (Figure 7D). ENST00000326094 negatively influenced six features in BRCA but positively influenced four features in PCPG (Figure 6C, Appendix A). ENST00000375991 negatively influenced more than four features in BLCA, BRCA, ESCA, and LGG but positively influenced PCPGs and TGCTs (Figure 6D, Appendix A).

To elucidate the mechanism of the impact of *ISCA1* on genomic heterogeneity, we analyzed the linear correlation between *ISCA1* and genomic heterogeneity. The results revealed that *ISCA1* RNA was positively correlated with at least four indicators in BLCA and LUSC but negatively correlated in BRCA (Figure 6E, Appendix A). *ISCA1* CNV was negatively correlated with more than four indicators in BRCA, CESC, LIHC, STAD, and UCEC (Appendix A). Additionally, *ISCA1* methylation was negatively correlated with four or more indicators in HNSC, LUAD, and TGCT (Appendix A). Furthermore, ENST00000311534 negatively affected more than four indicators in BRCA (Figure 6F, Appendix A), whereas ENST00000326094 had a positive effect on BLCA and BRCA (Figure 6G, Appendix A), and ENST00000375991 positively influenced BLCA and LUSC (Figure 6H, Appendix A).

### 3.6. Drug Prediction and Validation

On the basis of the correlations between the IC50 value and *ISCA1* RNA expression, 164 chemical compounds were identified as potential drugs that either promoted or inhibited *ISCA1* RNA expression (Appendix A). Appendix A displays the results of some of these chemical compounds. Three drugs that have undergone clinical trials and received FDA approval—pluripotin, chelerythrine, and fenretinide—were selected from these drugs (Figure 7A–C). The chemical structures of pluripotin and chelerythrine were obtained from the ZINC15 database, and we further analyzed the complete protein structure of ISCA1 to assess their potential interactions with these two drugs. The docking scores between ISCA1 and pluripotin and chelerythrine were found to be −25.22 and −22.05, respectively. Pluripotin was observed to interact with the Ser73 amino acid of ISCA1 through a hydrogen bond (Figure 7D), whereas chelerythrine is only bound to the surface of the ISCA1 protein (Figure 7E).

The complex system of chemical compounds and ISCA1 exhibited a certain level of stability. To further analyze the stability of the system formed by the compounds and ISCA1, we conducted molecular dynamics simulations. The Rg experiment was employed to describe the distribution characteristics of atoms along a specific axis and to assess the compactness of molecules. A smaller Rg value indicates a more stable protein structure. The values of the pluripotin and chelerythrine systems decreased, suggesting torsion of the ISCA1 protein during the simulation process. The ISCA1 protein was more stable in the ISCA1–chelerythrine system before 60 ns, whereas it was more stable in the ISCA1–chelerythrine system after 80 ns (Figure 7F).

Furthermore, the protein ISCA1 was more stable than the two protein–drug systems after 80 ns (Figure 7F). The RMSD experiment revealed that the compound and protein complex structure underwent a certain degree of displacement in position over time. The RMSD results indicated that the complex structure of chelerythrine and ISCA1 exhibited more stable fluctuations than did the ISCA1–pluripotin complex system and the ISCA1 protein after 50 ns (Figure 7G). Moreover, the RMSF experiment illustrated the distribution characteristics of the protein–compound complex system along a specific axis, with the molecules becoming more tightly packed as fluctuations decrease. The molecule was more tightly packed in the ISCA1–chelerythrine system than in the ISCA1–pluripotin system and ISCA1 protein (Figure 7H). The SASA experiment revealed that a smaller area on each residue corresponded to better system performance. The fluctuations in SASA performance in the two systems and the ISCA1 protein were similar (Figure 7I). On the basis of the RMSD and RMSF values obtained from the simulation experiments, the ISCA1–chelerythrine system was determined to be superior to the ISCA1–pluripotin system. We found that the number of hydrogen bonds in the two complex systems was less than that in the ISCA1 protein (Figure 7J).

### 3.7. The Comprehensive Analysis of the THCA

*ISCA1* could be a risk factor for ACC and THCA in OS and DSS; thus, we focused on THCA for subsequent research. THCA patients were divided into *ISCA1*-high and *ISCA1*-low tumors on the basis of *ISCA1* RNA expression and overall survival. The mutation frequencies of ferroptosis-related genes were compared between high- and low-*ISCA1* THCA patients. The genes with a mutation frequency exceeding 20% were *NRAS* (42.4%) and *HRAS* (18.5%). However, there was no difference in the mutation frequencies of FRGs between high- and low-grade tumors (Figure 8A). The RNA differential expression of FRGs was calculated between high- and low-grade tumors, with 55 and 96 FRGs identified as up- and downregulated, respectively, in high-grade tumors (Figure 8B, Appendix A). Subsequently, 273 biological process (BP), four cellular component (CC), and 56 molecular function (MF) terms of the upregulated FRGs (Appendix A), as well as 759 biological process, 13 cellular component, and 40 molecular function terms of the downregulated FRGs (Appendix A), were identified through GO annotation. The top five BP pathways of the up- and downregulated FRGs were the cellular response to metal ions, the cellular response to inorganic substances, the response to metal ions, the response to oxidative stress, and the fatty acid metabolic process (Appendix A), as well as the positive regulation of cytokine production, the regulation of the inflammatory response, the response to oxygen levels, the regulation of small molecule metabolic processes, and the response to hypoxia (Appendix A), respectively. Furthermore, 30 and 69 KEGG pathways were enriched in the upregulated and downregulated FRGs, respectively (Appendix A). GSVA was subsequently employed to investigate the enrichment of hallmark pathways in hot- and cold-treated tumors. Seventeen pathways, including allograft rejection, coagulation, and the interferon-γ response, were enriched in high-grade tumors, whereas 12 pathways, including pancreas β cells, heme metabolism, and bile acid metabolism, were enriched in cold-stage tumors (Figure 8C). The interactions between immune cells in the low group were more intense. For example, the correlations between M1-type macrophages and plasma cells, as well as follicular helper T cells, were greater in the low group than in the high group (Figure 8D). The network involving *ISCA1*, FRGs, IRGs, and ICGs was more complex in low-grade tumors than in high-grade tumors (Figure 8E,F, Appendix A).

Furthermore, the activities of multiple steps in the anticancer immune response were evaluated through the TIP database. The low tumors displayed higher activity scores in most steps, whereas only the scores of step six (recognition of cancer cells by T cells) were consistently lower in low tumors. The higher scores of step six and lower scores of step seven (activity of killing cancer cells) suggest that insufficient neoantigen recognition by T-cell receptors may be the key immunosuppression factor in *ISCA1*-high tumors despite increased T-cell recruitment (Figure 8G). Next, we calculated the correlations between the *ISCA1* score and activity score in high- and low-*ISCA1* tumors. We found that *ISCA1* RNA expression was negatively correlated with step two (cancer antigen presentation) expression in high-grade tumors but positively correlated with step 2 (cancer antigen presentation) expression in cold-stage tumors. The activity scores of step three (priming and activation), step four (B-cell recruitment and monocyte recruitment), and step five (infiltration of immune cells into tumors) were negatively related to *ISCA1* in low-grade tumors, but this correlation was lost in high-grade tumors. The correlation coefficient between step six and *ISCA1* was significantly greater in low-risk tumors (Figure 8H,I).

### 3.8. Single-Cell Analysis of ISCA1 in the THCA

The single-cell data from GSE191288 were utilized for further analysis. In terms of markers, eight cell types were identified, including follicular cells, T cells, pericyte cells, myeloid cells, endothelial cells, fibroblasts, B cells, and mast cells (Figure 9A,B). *ISCA1* was highly expressed in follicular cells, T cells, pericyte cells, myeloid cells, and fibroblasts (Figure 9C). The cells were then divided into *ISCA1*-activated and *ISCA1*-nonactivated cells within each cell type on the basis of *ISCA1* RNA expression. For example, T cells were categorized into *ISCA1*-activated and *ISCA1*-nonactivated T cells (Figure 9D). The percentage of the *ISCA1*-nonactivated group was greater in each cell type, particularly in mast cells, where the percentage of *ISCA1*-nonactivated cells exceeded 80% (Figure 9E). Differentially expressed genes between the activated and nonactivated cell groups were identified within each cell type (Appendix A). The GO and KEGG pathways of the up- and downregulated genes in each cell type were subsequently analyzed, and we observed that the primary BP and KEGG pathways varied among the different cell types. (Appendix A).

Additionally, the differentially expressed FRGs are displayed in Figure 9F, and seven and nine FRGs were upregulated in B cells and mast cells, respectively (Appendix A). We found that the upregulated FRGs in high THCA tumors in the TCGA database were not DEGs between B cells and mast cells (Figure 9F). These findings suggest that FRGs play different roles in different cell types. We also compared the scores for ferroptosis-driver genes, ferroptosis-suppressor genes, ferroptosis-marker genes, ferroptosis-unclassified regulator genes, and FRGs between *ISCA1*-activated and *ISCA1*-nonactivated cells in each cell type (Figure 9G, Appendix A). We detected significant differences in the scores of FRGs in each cell type, with higher scores in the *ISCA1*-activated group than in the control group, except for mast cells (Figure 9G). The scores of ferroptosis-driver genes, ferroptosis-marker genes, and ferroptosis-unclassified regulator genes were significantly greater in the activated groups than in the control groups in each cell type, except mast cells (Appendix A). *ISCA1* regulated the mechanism of FRGs in most cell types; these findings need further verification.

The CellChat package was used to infer cell–cell communication among different cell types in both the positive and negative cell groups. The number of inferred interactions was almost the same between the activated and nonactivated cell groups (Figure 10A,B, Appendix A). However, the interaction strength in the activated group was greater than that in the nonactivated group (Figure 10A,B, Appendix A). In the activated cell group, no other types of cells interacted with mast cells (Figure 10A,B), indicating a significant difference in the communication network between the two cell groups. The analysis of relative information flow revealed that the GH and PRL signaling pathways were specific to the activated cell group, whereas the AGT, NPR1, CD30, CD137, CD70, and CHEMERIN signaling pathways were specific to the nonactivated cell group (Figure 10C). Furthermore, our study demonstrated changes in the communication of potential ligand–receptor pairs between the two groups. In the activated group, T cells extensively communicated with pericyte cells, endothelial cells, fibroblasts, and mast cells through the CCL5-CCR5, CCL5-CCR3, CCL5-CCR1, CCL4-CCR5, and CCL11-CCR3 pathways. However, these interaction relationships were not detected in the nonactivated group (Figure 10D). In the activated group, myeloid cells interacted with other cells, with the exception of mast cells, through the CXCL3-CXCR2 pathway, which was not observed in the nonactivated group (Figure 10D).

## 4. Discussion

Many studies have indicated that the mechanisms of ferroptosis can impact the effectiveness of cancer treatment and resistance to cancer therapy [54]. Ferroptosis is particularly vulnerable in renal cell carcinoma, as shown by Yang et al., who reported that GPX4 regulates ferroptosis and inhibits tumor growth in mouse models [55]. Ubellacker et al. also revealed that melanoma cells tend to form more metastases through the lymphatic system than through the blood to evade ferroptosis [56]. Therefore, ferroptosis plays a crucial role in tumor suppression and can be utilized in cancer therapy. The accumulation of the cellular labile iron pool (LIP) resulting from the inhibition of divalent metal transporter 1 (DMT1) can trigger ferroptosis, leading to the elimination of breast cancer stem cells and the reversal of multidrug resistance [57]. Sun et al. demonstrated that inhibiting the expression of metallothionein-1G (MT-1G) can increase the anticancer activity of sorafenib through the induction of ferroptosis both in vitro and in vivo [58]. Additionally, the histone deacetylase inhibitor vorinostat can promote ferroptosis in epidermal growth factor receptor-tyrosine kinase inhibitor (EGFR)-mutant lung adenocarcinoma cells by suppressing the expression of solute carrier family 7 member 11 (SLC7A11) and improving the efficacy of ferroptosis inducers [59]. These findings suggest that inducing ferroptosis may help overcome immunotherapy resistance. We aimed to uncover the potential connection between *ISCA1* and ferroptosis in killing tumor cells across various types of cancer.

In this study, we conducted a comprehensive analysis to reveal potential relationships between *ISCA1* and FRGs across cancers. *ISCA1* served as a potential biomarker for ACC and THCA according to the survival analysis, with a focus on THCA for further analysis. *ISCA1* RNA expression was positively correlated with *SNX4* RNA expression, with a correlation coefficient of more than 0.8, whereas it was negatively correlated with the RNA expression of *ALOX5* and *MUC1,* with correlation coefficients of less than −0.4 in THCA (Appendix A). The correlation between the CNV and methylation of *ISCA1* and the RNA expression of FRGs was weak (Appendix A). The trafficking of growth factor receptors that can activate pro-oncogenic pathways might be regulated by sorting Nexin 4 (SNX4). Any dysregulation of this system could lead to aberrant signaling conducive to cancer development or progression [60]. The role of *SNX4* in the endocytic recycling of membrane proteins, including integrins and other adhesion molecules, can affect cell adhesion dynamics and migration capabilities [61]. By modulating signaling pathways through the lifecycle management of receptors and other signaling molecules, *SNX4* can indirectly influence apoptotic pathways and cell cycle regulators [62]. Some studies have demonstrated that arachidonate 5-lipoxygenase (ALOX5), and its metabolites can directly affect cancer cell dynamics by promoting proliferation or survival [63]. In cancer, *MUC1* is often dysregulated, leading to its overexpression, which significantly changes cellular behavior [64]. *MUC1* can mediate cell–cell and cell–matrix interactions, affecting adhesion properties and potentially facilitating detachment and metastasis [65]. *MUC1* interacts with several key signaling pathways, such as the NF-κB, β-catenin/Wnt, and PI3K/Akt pathways. These interactions can lead to increased cell proliferation, survival, and resistance to apoptosis [66]. *MUC1* can interfere with immune recognition and destruction of cancer cells [67]. Its overexpression may shield tumor cells from immune surveillance or create a local immunosuppressive microenvironment [68]. Therefore, we hypothesized that *ISCA1* affects THCA growth, possibly by influencing the expression of *SNX4*, *ALOX5*, and *MUC1*.

We then analyzed *ISCA1* and immune-related genes, including IRGs and ICGs. In THCA, *ISCA1* RNA expression was found to be positively correlated with *VEGFA* RNA expression, with a correlation coefficient of more than 0.5, and negatively correlated with *TNFRSF18* and *HLA-B* RNA expression, with correlation coefficients of less than −0.4 (Appendix A). The correlations between the CNV and methylation of *ISCA1* and the RNA expression of immune-related genes were weak (Appendix A). Additionally, *ISCA1* RNA expression was negatively associated with the infiltration of Tregs according to the CIBERSORT, QUANTISEQ, and xCell algorithms, with *p* values of less than 0.01 (Appendix A). Tregs play a crucial role in inhibiting the function of effector T cells, which are responsible for attacking and killing cancer cells [69]. They achieve this through various mechanisms, including the secretion of immunosuppressive cytokines such as IL-10 and TGF-β, consumption of local IL-2 (necessary for effector T-cell proliferation), and direct cell–cell interactions that inhibit effector T-cell activation [70]. By suppressing antitumor immunity, Tregs facilitate the growth and survival of tumors [71]. Therefore, *ISCA1* may inhibit the growth of THCA cancer cells by reducing Treg infiltration.

The regulatory network involving *ISCA1*, FRGs, IRGs, and ICGs was more complex in the low-*ISCA1*-expressing group. In this group, higher scores were observed for step six (recognition of cancer cells by T cells), whereas lower scores were noted for step seven (activity of killing cancer cells). These findings suggest that a regulatory relationship between *ISCA1* and FRGs may aid T cells in recognizing cancer cells but may not effectively kill them. Therefore, our research focused on identifying ways to promote the expression of *ISCA1* and effectively recognize the antigen.

In the drug prediction study, 164 small molecules, including chelerythrine and fenretinide, were identified to have IC50 values that correlated positively with *ISCA1* RNA expression. Chelerythrine acts as a potent PKC inhibitor and has the potential to inhibit the proliferation of cancer cells and induce apoptosis [72]. Its anti-inflammatory properties could also enhance its anticancer effects [73]. Evidence suggests that chelerythrine can inhibit angiogenesis, a crucial process for tumor growth and metastasis [74]. Fenretinide promotes apoptosis in cancer cells through receptor-mediated and mitochondrial apoptotic pathways. It increases the levels of ROS within cells, leading to oxidative stress and triggering apoptosis [75]. By inhibiting angiogenesis, fenretinide can hinder the growth of tumors by depriving them of necessary nutrients [76]. However, chelerythrine may also affect healthy cells, potentially leading to toxicity [77]. To address this issue, combination therapy with other drugs should be considered to reduce toxicity. Future research will focus on investigating the synergistic and antagonistic effects of different small molecules [78].

## 5. Conclusions

In conclusion, our study revealed a significant relationship between *ISCA1* and ferroptosis-related genes, immune regulatory genes, immune checkpoint genes, the immune microenvironment, tumor stemness, and genomic heterogeneity across cancers. Furthermore, *ISCA1* could serve as a valuable biomarker for predicting prognosis and assessing the effectiveness of immunotherapy in THCA patients. *ISCA1* inhibits the proliferation of THCA cancer through the interaction of TRGs with the immune microenvironment. However, further exploration and verification through additional basic experiments and clinical trials are necessary to elucidate the precise molecular mechanism underlying *ISCA1*-mediated functions in tumorigenesis and immunotherapy of THCA. These future endeavors will increase our understanding of the role of *ISCA1* and its potential as a therapeutic target in the treatment of THCA.

## Figures and Tables

**Figure 1 genes-15-01538-f001:**
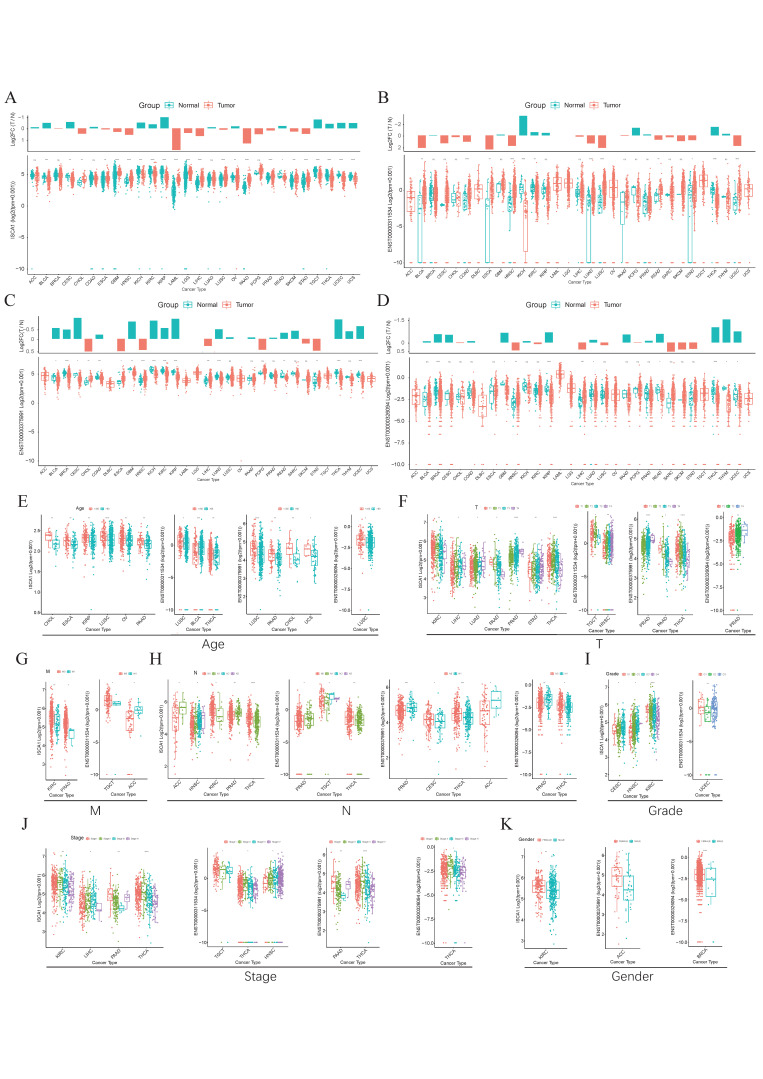
The expression levels of *ISCA1* in various cancers. (**A**) Differential expression of (**A**) *ISCA1*, (**B**) ENST00000311534, (**C**) ENST00000375991, and (**D**) ENST00000326094 between various normal and cancer tissues. Expression of *ISCA1*, ENST00000311534, ENST00000375991, and ENST00000326094 between (**E**) age groups, (**F**) T groups, (**G**) M groups, (**H**) N groups, (**I**) grade groups, (**J**) stage groups, and (**K**) sex groups.

**Figure 2 genes-15-01538-f002:**
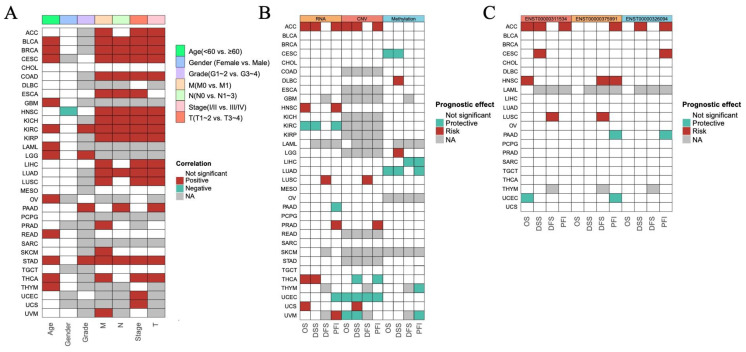
Predictive effects of *ISCA1* expression levels on prognosis in human cancers. (**A**) Heatmap clusters showing the correlation between *ISCA1* expression levels and overall survival (OS) according to seven clinical characteristics of 33 cancer types. (**B**) Heatmap showing the correlation between *ISCA1* expression levels and four curated survival outcomes, including overall survival (OS), disease-specific survival (DSS), the disease-free interval (DFI), and the progression-free interval (PFI). (**C**) Heatmap showing the correlation between the levels of three transcripts of *ISCA1* (ENST00000311534, ENST00000326094, and ENST00000375991) and four curated survival outcomes. Survival analysis was performed via univariate Cox regression on the basis of curated survival data from the TCGA database. Red boxes represent a risk factor, green boxes represent a protective factor, white boxes represent that the analyses are not significant, and gray boxes represent that the data are not available).

**Figure 3 genes-15-01538-f003:**
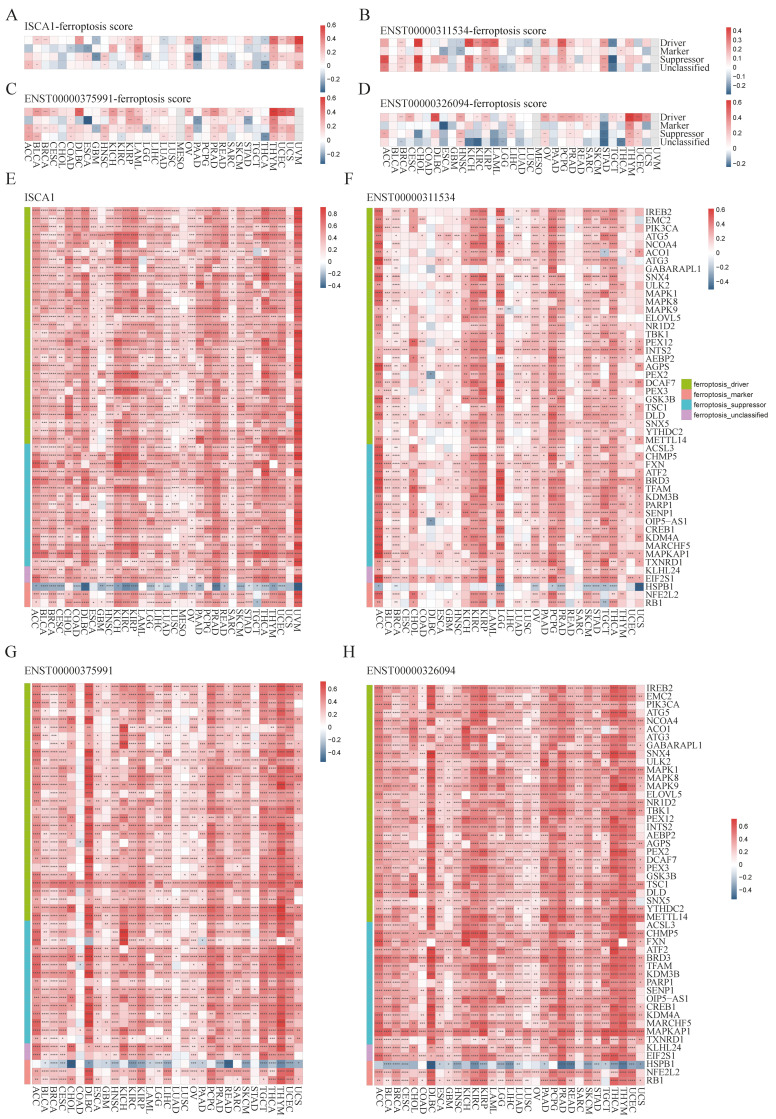
Multiomics analysis of *ISCA1* and ferroptosis-related genes in each cancer type. The heatmap illustrates the correlations between (**A**) *ISCA1*, (**B**) ENST00000311534, (**C**) ENST00000375991, (**D**) ENST00000326094, and ferroptosis-related gene set scores. The heatmap shows the correlations between (**E**) *ISCA1*, (**F**) ENST00000311534, (**G**) ENST00000375991, and (**H**) ENST00000326094 and ferroptosis-related genes (*: *p* < 0.05, **: *p* < 0.01, ***: *p* < 0.001, ****: *p* < 0.0001).

**Figure 4 genes-15-01538-f004:**
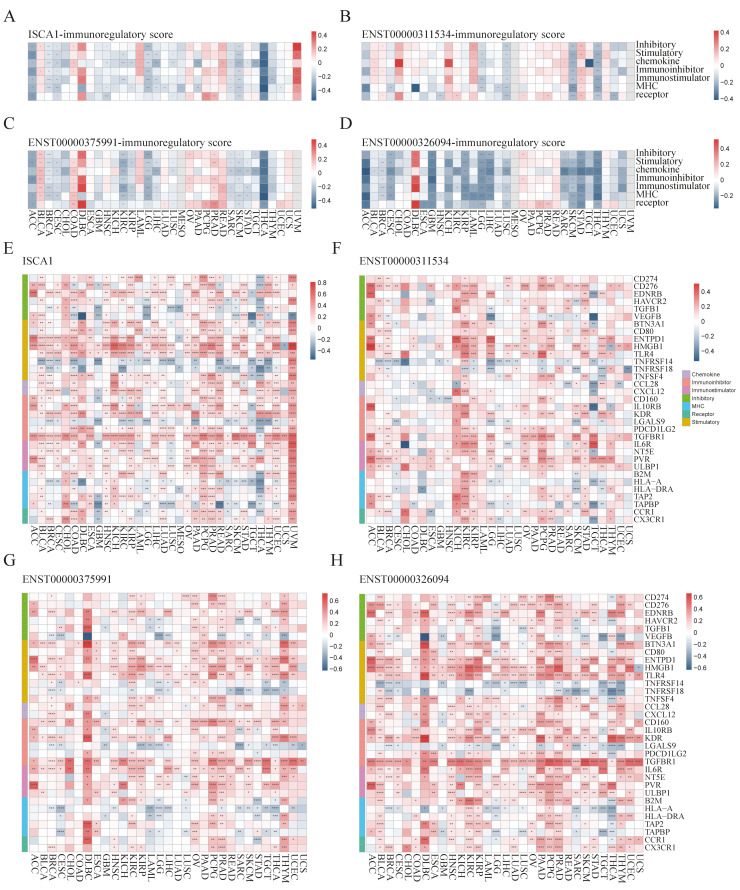
Multiomics analysis of *ISCA1* and immunoregulatory genes in each cancer. The heatmap illustrates the correlations between (**A**) *ISCA1*, (**B**) ENST00000311534, (**C**) ENST00000375991, (**D**) ENST00000326094, and immunoregulatory-related gene set scores. The heatmap shows the correlations between (**E**) *ISCA1*, (**F**) ENST00000311534, (**G**) ENST00000375991, (**H**) ENST00000326094, and immunoregulatory-related genes (*: *p* < 0.05, **: *p* < 0.01, ***: *p* < 0.001, ****: *p* < 0.0001).

**Figure 5 genes-15-01538-f005:**
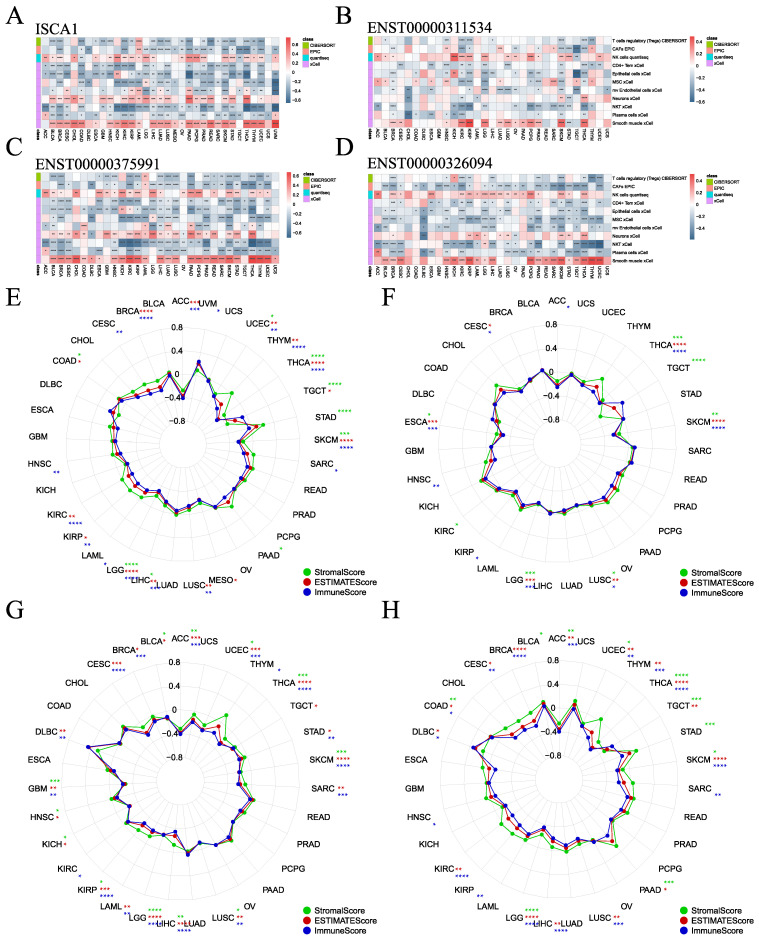
Association of *ISCA1* with immune infiltration across cancers. (**A**) Heatmaps illustrating the correlation between (**A**) *ISCA1*, (**B**) ENST00000311534, (**C**) ENST00000375991, (**D**) ENST00000326094 RNA expression, and immune cell infiltration according to the six algorithms. Correlation analysis between (**E**) *ISCA1*, (**F**) ENST00000311534, (**G**) ENST00000375991, and (**H**) ENST00000326094 RNA expression and three immune infiltration scores (*: *p* < 0.05, **: *p* < 0.01, ***: *p* < 0.001, ****: *p* < 0.0001).

**Figure 6 genes-15-01538-f006:**
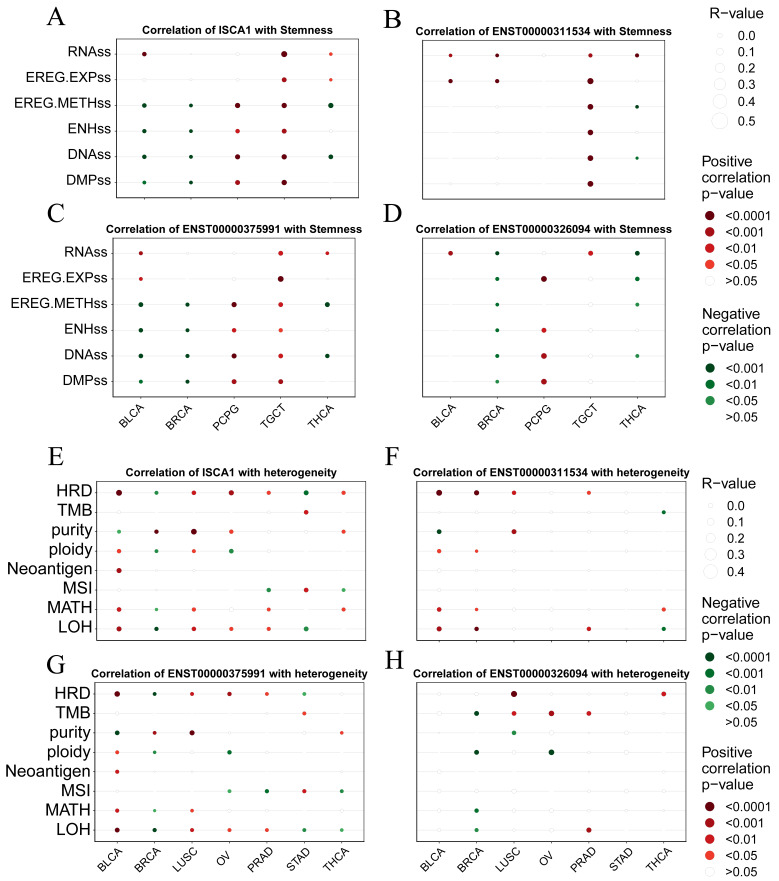
Correlation analysis between *ISCA1* expression and tumor stemness and genomic heterogeneity in each cancer. Bubble plots illustrating the correlation between (**A**) *ISCA1*, (**B**) ENST00000311534, (**C**) ENST00000375991, (**D**) ENST00000326094 RNA expression, and tumor stemness. Bubble plots illustrating the correlation between (**E**) *ISCA1*, (**F**) ENST00000311534, (**G**) ENST00000375991, and (**H**) ENST00000326094 RNA expression and genomic heterogeneity. The bubble size represents the strength of the correlation, and the bubble color indicates significance, with red indicating a positive correlation and green indicating a negative correlation.

**Figure 7 genes-15-01538-f007:**
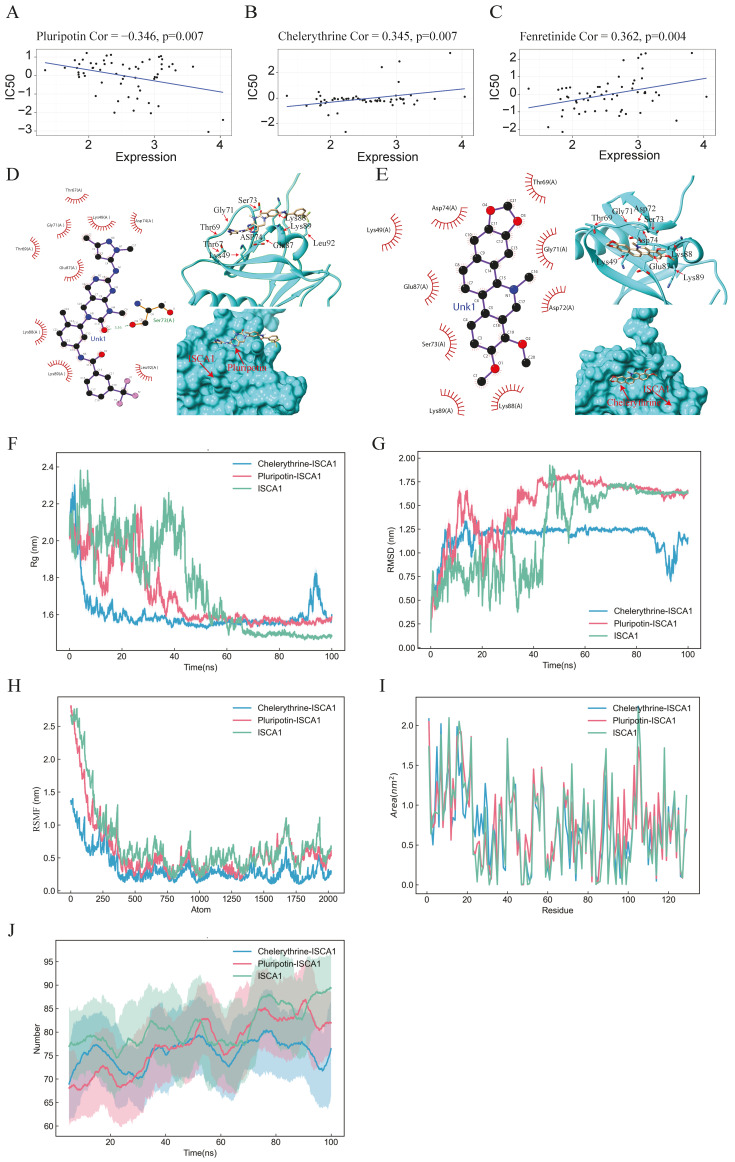
Drug prediction, molecular docking, and molecular dynamics analysis. (**A**–**C**) Scatter plots illustrating the correlation between *ISCA1* expression and the sensitivity of cells to pluripotin, fenretinide, and chelerythrine. (**D**,**E**) 2D and 3D docking diagrams depicting the molecular docking interactions between ISCA1 and pluripotin, as well as between ISCA1 and chelerythrine. (**F**,**G**) Rg and RMSD values of the ISCA1–pluripotin and ISCA1–chelerythrine systems as well as the ISCA1 protein over time. (**H**) RMSF values of the two systems and the ISCA1 protein versus the number of atoms. (**I**) SASA values of two systems and the ISCA1 protein over each residue number. (**J**) The number of hydrogen bonds in the two systems and the ISCA1 protein over time.

**Figure 8 genes-15-01538-f008:**
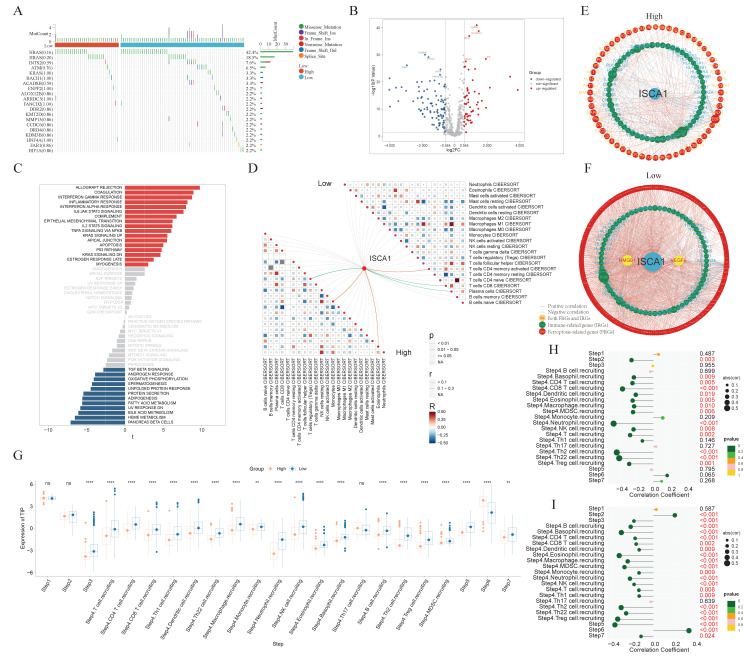
The comprehensive analysis of the THCA. (**A**) Waterfall plot showing the difference in somatic mutation frequency between *ISCA1*-high and *ISCA1*-low tumors. (**B**) Differentially expressed FRGs between high- and low-grade tumors. (**C**) Different hallmark pathways between high- and low-grade tumors. The networks of *ISCA1*, FRGs, and immunoregulation genes in (**D**) high- and (**E**) low-grade tumors. (**F**) Correlations between immune cells in high- and low-grade tumors. (**G**) Comparison of the TIP scores of high- and low-grade tumors. Correlations between *ISCA1* RNA expression and TIP scores in (**H**) high and (**I**) low tumors (*: *p* < 0.05, **: *p* < 0.01, ***: *p* < 0.001, ****: *p* < 0.0001).

**Figure 9 genes-15-01538-f009:**
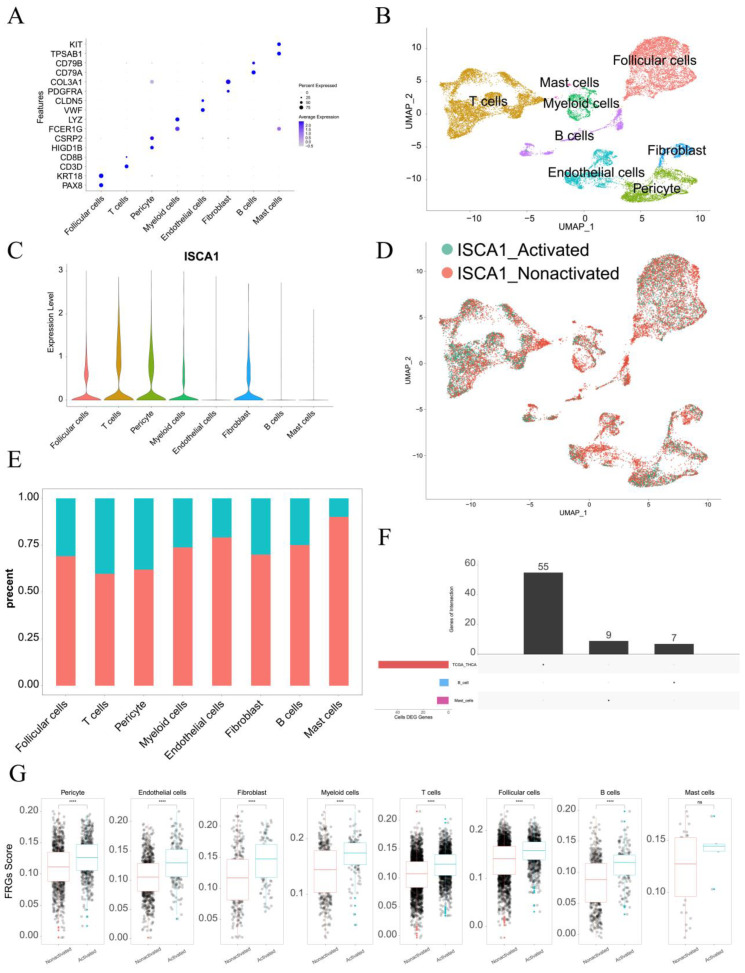
Single-cell analysis of *ISCA1* in the THCA. (**A**) Identification of cell types on the basis of marker genes in THCA. (**B**) UMAP plot showing the cells. (**C**) *ISCA1* RNA expression in different cell types. (**D**) UMAP plot showing *ISCA1*-activated and *ISCA1*-nonactivated cells on the basis of *ISCA1* RNA expression. (**E**) Percentages of activated and nonactivated cells in each cell type. (**F**) UpSet plot showing the intersection between the differentially expressed FRGs in each cell type and those in the TCGA database. (**G**) Comparison of the different scores of FRGs between the activated and nonactivated groups in each cell type (*: *p* < 0.05, **: *p* < 0.01, ***: *p* < 0.001, ****: *p* < 0.0001). ns: *p* ≥ 0.05.

**Figure 10 genes-15-01538-f010:**
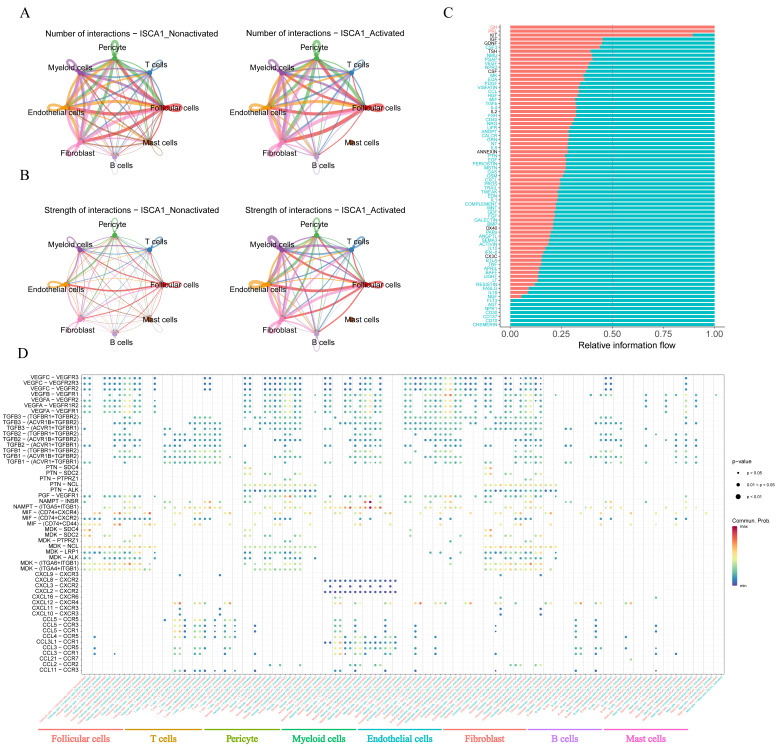
Cell–cell communication. (**A**) The number of interactions between different cell types in the activated and nonactivated groups. (**B**) The strength of interactions between different cell types in the activated and nonactivated groups. (**C**) The relative information flow in the activated and nonactivated groups. (**D**) Comparison of specific ligand–receptor interactions among cell types between the *ISCA1*-high group and the *ISCA1*-low group.

## Data Availability

The mRNA expression profile, mutation annotation, CNV, and clinical metadata data were obtained from the TCGA and GTEx databases. The protein expression data were downloaded from the CPTAC database. ScRNA-seq data were obtained from GSE191288. The full code used during the current study is available at https://github.com/sangmm12/ISCA1_pancancer.

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
