# Peer review of "Comprehensive Analysis Reveals That ISCA1 Is Correlated with Ferroptosis-Related Genes Across Cancers and Is a Biomarker in Thyroid Carcinoma"

_genes, 2024, doi:10.3390/genes15121538_

Round 1
Reviewer 1 Report
Comments and Suggestions for Authors
Xiong et al. compiled an analysis of the ISCA1 gene at pan cancer level and correlated its expression with markers of ferroptosis and immune infiltration, as well as drug predictions. The idea and design of this work is interesting, however the manuscript is extremely long in terms of results and should be restructured to focus on the most important data. Therefore, I cannot provide a in-depth point-to-point review until the manuscript is comprehensively revised.
Major points:
- The expression overview in Figure 1 is extremely busy, but could be reduced to only show expression differences of the 3 transcript variants in normal and tumour tissues. These are shown in Figure 2, but the expression levels between different clinicopathological features was not performed for each transcript variant. I suggest to significantly reduce data from Figure 1 and 2 and combine into a single Figure, streamlining the findings.
- The authors show the expression changes between normal and tumour tissue, presented as boxplots in Fig 1C. However, there is no description of these results, except the downregulation in THCA, KIRC, and PAAD. However, these are not the only tumours where ISCA1 is downregulated. It would be beneficial to show a fold change plot to summarise what cancers show up-regulation or down-regulation of the gene.
- Most of the plots are not described in the Results paragraph 3.1, such as the different expression levels for age and gender (Fig 1 D-E). These figures could be moved to supplementary or better explained in the results. In line 222-224, this statement cannot be made from the data in Fig 1F and 1I, as there are no markers of tumour heterogeneity and malignancy. The authors compare different metrics of tumour progression in plots 1F-J, however they are not sufficiently explained in results nor in the figure legend.
- Because of the comprehensive analysis for THCA, I suggest that the authors reduce the data load at pan-cancer level. For instance, the section "3.6 Single-cell expression levels of ISCA1 in pan-cancers" does not add additional insights to the previous analysis and could be omitted.
- The drug prediction and molecular dynamics is interesting. Because it is based on expression levels, I would suggest to reduce the rest of the manuscript by only focusing on expression levels, and reducing the data load by omitting methylation and CNV status. While interesting, these could be summarised as supplementary information. This would help focusing Fig 6 and 7.
- Figures 10 and 11 are very busy, but represent a crucial part of the study. I suggest that these figures are rearranged and distributed to additional figures, by reducing the rest of the analysis at pancan level.
Author Response
1、The expression overview in Figure 1 is extremely busy, but could be reduced to only show expression differences of the 3 transcript variants in normal and tumour tissues. These are shown in Figure 2, but the expression levels between different clinicopathological features was not performed for each transcript variant. I suggest to significantly reduce data from Figure 1 and 2 and combine into a single Figure, streamlining the findings.
Response:
We appreciate the reviewer for providing us with helpful feedback. In this study, we analyzed the expression differences of the three transcript variants in normal and tumor tissues, as shown in Figures 1B-D. Additionally, we examined the expression differences of the three transcript variants among different clinicopathological features, as presented in Figures 1G-K. We have now combined Figures 1 and 2 into a single figure for clarity.
2、The authors show the expression changes between normal and tumour tissue, presented as boxplots in Fig 1C. However, there is no description of these results, except the downregulation in THCA, KIRC, and PAAD. However, these are not the only tumours where ISCA1 is downregulated. It would be beneficial to show a fold change plot to summarise what cancers show up-regulation or down-regulation of the gene.
Response:
We appreciate the reviewer for providing helpful feedback. In response, we have expanded on the results in detail in lines 253-256. Additionally, we have included fold change plots in Figure 1A-D to summarize which cancers exhibit upregulation or downregulation of genes or transcripts.
3、Most of the plots are not described in the Results paragraph 3.1, such as the different expression levels for age and gender (Fig 1 D-E). These figures could be moved to supplementary or better explained in the results. In line 222-224, this statement cannot be made from the data in Fig 1F and 1I, as there are no markers of tumour heterogeneity and malignancy. The authors compare different metrics of tumour progression in plots 1F-J, however they are not sufficiently explained in results nor in the figure legend.
Response:
We want to express our gratitude to the reviewer for providing valuable feedback. In response to their suggestions, we have elaborated on the various expression levels for age and sex, as described in lines 261-275. Additionally, we have moved the statement to “In addition, advanced patients presented increased levels of tumor heterogeneity and malignancy due to low expression of ISCA1” in this revision. Furthermore, we have adequately explained in the Results section the different metrics of tumor progression, as shown in Figures 1E-K, as well as in lines 261-275.
4、Because of the comprehensive analysis for THCA, I suggest that the authors reduce the data load at pan-cancer level. For instance, the section "3.6 Single-cell expression levels of ISCA1 in pan-cancers" does not add additional insights to the previous analysis and could be omitted.
Response:
We thank the reviewer for their helpful feedback. We have removed the results of “Single-cell expression levels of ISCA1 across cancers” in this revised version.
5、The drug prediction and molecular dynamics is interesting. Because it is based on expression levels, I would suggest to reduce the rest of the manuscript by only focusing on expression levels, and reducing the data load by omitting methylation and CNV status. While interesting, these could be summarised as supplementary information. This would help focusing Fig 6 and 7.
Response:
We thank the reviewer for their helpful feedback. In this revision, we have attached the CNV and methylation results as supplementary figures.
6、Figures 10 and 11 are very busy, but represent a crucial part of the study. I suggest that these figures are rearranged and distributed to additional figures, by reducing the rest of the analysis at pancan level.
Response:
We thank the reviewer for their helpful feedback. In this revision, we have divided the original Figure 11 into two new figures, now referred to as Figure 9 and Figure 10.
Reviewer 2 Report
Comments and Suggestions for Authors
Comments and Suggestions:
Title: Comprehensive analysis reveals ISCA1 correlated with ferroptosis-related genes in pan-cancer and as a biomarker in thyroid carcinoma
In this manuscript Xiong and colleagues performed extensive bioinformatics analyses in which they presented the role of ISCA1 protein in tumor progression by interacting with ferroptosis related genes and tumor immune microenvironment using several tools and databases. They identified drugs including pluripotin, chelerythrine, and fenretinide that could inhibit or promote the ISCA1 gene expression. They concluded that ISCA1 could serve as a potential biomarker for predicting prognosis and immune therapeutic efficacy in thyroid carcinoma.
Prognostic value of SLCA1 genes associated with ferroptosis has been reported in other malignancies previously and the results presented herein are in line with findings in other tumor types. The analyses are extensive and thorough and the results may be clinically useful.
Major Points:
1. Figure 10B and line 183: The differential gene analysis was done using pvalue<0.05 and a log fold change (logFC) greater than 0.25 or less than -0.25 were identified as DEGs. The logFC cutoff values in the line 183 differs from Figure 10B. The Fold Change cutoff should be taken as either 2 (logFC=±1) or 1.5 (logFC=±0.584). The values of lofFC cutoff taken in this is very less for considering a gene to be deregulated. Please re-analyze the data.
2. The in-silico study is not enough to prove that the three drugs inhibits or promote SLCA1 gene expression. The validation study of binding of these drugs pluripotin, chelerythrine, and fenretinide with the SLCA1 protein could be done in wet lab also.
Minor Points:
1. Figure 11: the number of sub-figures is more and difficult to read and understand. Please split the figure into two or more figures or submit as supplementary figures.
2. Line 424: The labels ‘G-I’ should be corrected as ‘J-L’.
3. Please cite the figure 10F in the text.
Author Response
Reviewer #2:
Title: Comprehensive analysis reveals ISCA1 correlated with ferroptosis-related genes in pan-cancer and as a biomarker in thyroid carcinoma
In this manuscript Xiong and colleagues performed extensive bioinformatics analyses in which they presented the role of ISCA1 protein in tumor progression by interacting with ferroptosis related genes and tumor immune microenvironment using several tools and databases. They identified drugs including pluripotin, chelerythrine, and fenretinide that could inhibit or promote the ISCA1 gene expression. They concluded that ISCA1 could serve as a potential biomarker for predicting prognosis and immune therapeutic efficacy in thyroid carcinoma.
Prognostic value of SLCA1 genes associated with ferroptosis has been reported in other malignancies previously and the results presented herein are in line with findings in other tumor types. The analyses are extensive and thorough and the results may be clinically useful.
Major Points:
1、Figure 10B and line 183: The differential gene analysis was done using pvalue<0.05 and a log fold change (logFC) greater than 0.25 or less than -0.25 were identified as DEGs. The logFC cutoff values in the line 183 differs from Figure 10B. The Fold Change cutoff should be taken as either 2 (logFC=±1) or 1.5 (logFC=±0.584). The values of lofFC cutoff taken in this is very less for considering a gene to be deregulated. Please re-analyze the data.
Response:
We thank the reviewer for their helpful feedback. The log2FC cutoff value was ±0.584 in this revision. We reanalyzed the data, and the results are shown in Figure 8B and Supplementary Figure 18.
2、The in-silico study is not enough to prove that the three drugs inhibits or promote SLCA1 gene expression. The validation study of binding of these drugs pluripotin, chelerythrine, and fenretinide with the SLCA1 protein could be done in wet lab also.
Response:
We appreciate the reviewer's suggestions and completely agree with their comments regarding the importance of incorporating in vitro and in vivo data to strengthen this study. In fact, we are currently planning a separate comprehensive study on ISCA1 function, where we will conduct numerous in vivo and in vitro experiments to demonstrate the effects of the drugs pluripotin, chelerythrine, and fenretinide on ISCA1 gene expression.
Minor Points:
1、Figure 11: the number of sub-figures is more and difficult to read and understand. Please split the figure into two or more figures or submit as supplementary figures.
Response:
We thank the reviewer for their helpful feedback. In this revised version, we divided the original Figure 11 into two separate figures, now referred to as Figure 9 and Figure 10.
2、Line 424: The labels ‘G-I’ should be corrected as ‘J-L’.
Response:
We thank the reviewer for their helpful feedback. In this revised version, we have rectified the error noted.
3、Please cite the figure 10F in the text.
Response:
We thank the reviewer for their helpful feedback. In this revision, we have changed the original Figure 10F to Figure 8F and have cited Figure 8F on line 554.
Reviewer 3 Report
Comments and Suggestions for Authors
Manuscript Title: “Comprehensive analysis reveals ISCA1 correlated with ferroptosis-related genes in pan-cancer and as a biomarker in thyroid carcinoma”
I have carefully reviewed the manuscript examining the role of ISCA1 in cancer development and immune regulation. This is an impressively comprehensive study that combines multi-omics analysis with sophisticated computational approaches to elucidate ISCA1's functions across various cancer types. I have to say that I am impressed by all the analyses conducted by the authors.
Major points
I have concerns regarding the statistical methodology described in paragraph 3.2, Specifically, the correlation analysis between expression levels and overall survival. A more appropriate approach would be to employ Cox proportional hazards regression analysis, which would provide hazard ratios and confidence intervals. Also, I would recommend including a Forest Plot, as this would clearly display the hazard ratios across different variables and subgroups while showing their respective confidence intervals.
I would like to suggest an alternative approach for analysing (paragraph 3.3) the relationship between ISCA1 and your gene sets of interest (for example,ferroptosis-related and immune checkpoint genes). Rather than examining individual gene correlations, implementing a gene set scoring method would provide more robust and biologically meaningful results. Specifically, I would suggest methods like single-sample Gene Set Enrichment Analysis (ssGSEA), singscore, or mean Z-score would allow you to generate a comprehensive score for each sample that reflects the collective behavior of the gene sets. This approach would better capture the biological variability, rather than relying on individual gene correlations.
Figure 6 and the heatmaps show too much information, and it is confusing; please try alternative ways to present the data. Maybe, just a simple filter in order show to the most significant results could be sufficient.
Figure 7 and the bubble plots show too much information, and it is confusing; as Figure 6, please try alternative ways to present the data.
Figure 11 is too complex, please try to simplify it (maybe some plots in supplementary).
The author should revise this sentence “Furthermore, drug prediction, molecular docking, and molecular dynamics analysis identified potential drugs, including pluripotin, chelerythrine, and fenretinide, that could promote or inhibit ISCA1 RNA expression levels” as follows “Furthermore, drug prediction and validation through molecular docking, and molecular dynamics analysis of identified potential drugs, including pluripotin, chelerythrine, and fenretinide, that could promote or inhibit ISCA1 RNA expression levels” in the abstract section.
Please also include essential dynamics and hydrogen bond analysis.
The author conducted a 50 ns MD simulation; however, the results shown in the graph suggest that a minimum of 100 ns is needed. Please extend the MD simulation to 100 ns.
The author should also perform an MD simulation of the free protein (protein alone) to compare these results with those of the protein-ligand complexes.
Minor points
Since there are a lot of figures and plots in the manuscript, I suggest to keep in the manuscripts only the relevant ones. In other words, maybe it is better to move the results regarding trascripts/proteins/... that are not changing to the supplementary.
A lot of figures and plots have different style and color. Please try to uniform them.
Figure 4 and Figure 5 figure legends are not visible in my version of the manuscript.
In the introduction section last paragraph “We also performed drug prediction, molecular docking and molecular dynamics analysis to search for candidate compounds that could promote or inhibit ISCA1 RNA expression” should rewrite as “We also performed drug prediction and validation through molecular docking and molecular dynamics analysis to identify candidate drugs that could promote or inhibit ISCA1 RNA expression”.
In the Materials and Methods as well as Results section, section 2.6 and 3.7, titled “Drug prediction, molecular docking and molecular dynamics analysis,” should revise as “Drug prediction and validation”.
The author analysed the molecular dynamics simulation results using RMSD, RMSF, Rg, and SASA;
The author selected three drugs-pluripotin, chelerythrine, and fenretinide-that have undergone clinical trials and received FDA approval. However, only two of these, pluripotin and chelerythrine, were validated through MD simulation. Please provide possible reasons for the exclusion of fenretinide.
Please clarify these two ideas (1) “Based on the correlations between the IC50 value and ISCA1 RNA expression, 164 chemical compounds were identified as potential drugs that either promoted or inhibited ISCA1 RNA expression” in the Results section and (2) “In the drug prediction study, 142 small molecules, including chelerythrine and fenretinide, were identified to have IC50 values that correlated positively with ISCA1 RNA expression” in the Discussion section.
Raw 123: CIBERSOR should be correct to CIBERSORT.
Raw 228 and Row 280: why NUSAP1 gene is reported? Is It ISCA1 instead?
Author Response
Reviewer #3:
Manuscript Title: “Comprehensive analysis reveals ISCA1 correlated with ferroptosis-related genes in pan-cancer and as a biomarker in thyroid carcinoma”
I have carefully reviewed the manuscript examining the role of ISCA1 in cancer development and immune regulation. This is an impressively comprehensive study that combines multi-omics analysis with sophisticated computational approaches to elucidate ISCA1's functions across various cancer types. I have to say that I am impressed by all the analyses conducted by the authors.
Major points
I have concerns regarding the statistical methodology described in paragraph 3.2, Specifically, the correlation analysis between expression levels and overall survival. A more appropriate approach would be to employ Cox proportional hazards regression analysis, which would provide hazard ratios and confidence intervals. Also, I would recommend including a Forest Plot, as this would clearly display the hazard ratios across different variables and subgroups while showing their respective confidence intervals.
Response:
We are grateful to the reviewer for their insightful feedback. When extracting genes, we classified those with a hazard ratio (HR) greater than 1 as risk factors and those with an HR less than 1 as protective factors. These results are illustrated in Figure 2. Additionally, forest plots were used to present hazard ratios and confidence intervals, which can be found in Supplementary Figures 3-8.
I would like to suggest an alternative approach for analysing (paragraph 3.3) the relationship between ISCA1 and your gene sets of interest (for example,ferroptosis-related and immune checkpoint genes). Rather than examining individual gene correlations, implementing a gene set scoring method would provide more robust and biologically meaningful results. Specifically, I would suggest methods like single-sample Gene Set Enrichment Analysis (ssGSEA), singscore, or mean Z-score would allow you to generate a comprehensive score for each sample that reflects the collective behavior of the gene sets. This approach would better capture the biological variability, rather than relying on individual gene correlations.
Response:
We thank the reviewer for their helpful feedback. The ssGSEA algorithm was used to calculate the gene set scores for ferroptosis driver genes, ferroptosis marker genes, ferroptosis suppressor genes, and ferroptosis-unclassified genes that are inhibitory, stimulatory, chemokine, immunoinhibitor, immunostimulator, MHC, and receptor genes.
Figure 6 and the heatmaps show too much information, and it is confusing; please try alternative ways to present the data. Maybe, just a simple filter in order show to the most significant results could be sufficient.
Response:
We thank the reviewer for their helpful feedback. We have simplified the results of Figure 6 (now referred to as Figure 5).
Figure 7 and the bubble plots show too much information, and it is confusing; as Figure 6, please try alternative ways to present the data.
Response:
We thank the reviewer for their helpful feedback. We have simplified the original Figure 6 and Figure 7; in the revised version, they are referred to as Figure 5 and Figure 6.
Figure 11 is too complex, please try to simplify it (maybe some plots in supplementary).
Response:
We thank the reviewer for their helpful feedback. In this revision, we divided the original Figure 11 into two separate figures, now defined as Figure 9 and Figure 10.
The author should revise this sentence “Furthermore, drug prediction, molecular docking, and molecular dynamics analysis identified potential drugs, including pluripotin, chelerythrine, and fenretinide, that could promote or inhibit ISCA1 RNA expression levels” as follows “Furthermore, drug prediction and validation through molecular docking, and molecular dynamics analysis of identified potential drugs, including pluripotin, chelerythrine, and fenretinide, that could promote or inhibit ISCA1 RNA expression levels” in the abstract section.
Response:
We thank the reviewer for their helpful feedback. In this revision, we have rewritten “We performed drug prediction and validation through molecular docking and molecular dynamics analysis to identify candidate drugs that could promote or inhibit ISCA1 RNA expression” in the abstract section, lines 40-42.
Please also include essential dynamics and hydrogen bond analysis.
Response:
We thank the reviewer for their helpful feedback. We performed this analysis, and the results are shown in Figure 7J.
The author conducted a 50 ns MD simulation; however, the results shown in the graph suggest that a minimum of 100 ns is needed. Please extend the MD simulation to 100 ns.
Response:
We thank the reviewer for their helpful feedback. We extended the MD simulation to 100 ns and present the results in Figures 7F-I.
The author should also perform an MD simulation of the free protein (protein alone) to compare these results with those of the protein-ligand complexes.
Response:
We thank the reviewer for their helpful feedback. We performed this MD simulation, and the results are shown in Figures 7F-I.
Minor points
Since there are a lot of figures and plots in the manuscript, I suggest to keep in the manuscripts only the relevant ones. In other words, maybe it is better to move the results regarding trascripts/proteins/... that are not changing to the supplementary.
Response:
We thank the reviewer for their helpful feedback. In this revision, we have attached the CNV and methylation results as supplementary figures.
A lot of figures and plots have different style and color. Please try to uniform them.
Response:
We thank the reviewer for their helpful feedback. The color and style of the article are different because we aim to create a more diverse range of graphics.
Figure 4 and Figure 5 figure legends are not visible in my version of the manuscript.
Response:
We thank the reviewer for their helpful feedback. We ensured that all the figure legends are visible in this revision.
In the introduction section last paragraph “We also performed drug prediction, molecular docking and molecular dynamics analysis to search for candidate compounds that could promote or inhibit ISCA1 RNA expression” should rewrite as “We also performed drug prediction and validation through molecular docking and molecular dynamics analysis to identify candidate drugs that could promote or inhibit ISCA1 RNA expression”.
Response:
We thank the reviewer for their helpful feedback. We have rewritten lines 40-42.
In the Materials and Methods as well as Results section, section 2.6 and 3.7, titled “Drug prediction, molecular docking and molecular dynamics analysis,” should revise as “Drug prediction and validation”.
Response:
We thank the reviewer for their helpful feedback. We titled sections 2.6 and 3.6 “Drug prediction and validation” in lines 172 and 488, respectively.
The author analysed the molecular dynamics simulation results using RMSD, RMSF, Rg, and SASA;
The author selected three drugs-pluripotin, chelerythrine, and fenretinide-that have undergone clinical trials and received FDA approval. However, only two of these, pluripotin and chelerythrine, were validated through MD simulation. Please provide possible reasons for the exclusion of fenretinide.
Response:
When performing molecular docking between ISCA1 and fenretinide, the following error was encountered: "Could not complete growth. Confirm that the grid box is large enough to contain the ligand." We tried to increase the volume of the grid box, but the error still persisted, causing the proteins and molecules to not dock successfully.
Please clarify these two ideas (1) “Based on the correlations between the IC50 value and ISCA1 RNA expression, 164 chemical compounds were identified as potential drugs that either promoted or inhibited ISCA1 RNA expression” in the Results section and (2) “In the drug prediction study, 142 small molecules, including chelerythrine and fenretinide, were identified to have IC50 values that correlated positively with ISCA1 RNA expression” in the Discussion section.
Response:
We thank the reviewer for their helpful feedback. A total of 164 chemical compounds were identified, and the chemical compounds are listed in Supplemental Table 1. We have corrected this error on line 692.
Raw 123: CIBERSOR should be correct to CIBERSORT.
Response:
We thank the reviewer for their helpful feedback. We have corrected this error on line 148.
Raw 228 and Row 280: why NUSAP1 gene is reported? Is It ISCA1 instead?
Response:
We thank the reviewer for their helpful feedback. We have corrected this error on lines 1053 and 1086.
Round 2
Reviewer 3 Report
Comments and Suggestions for Authors
I appreciate the author's substantial improvements to the document. However, I cannot fully endorse this version due to significant visual presentation issues. The graphical elements oe text still lack clarity, and the typography presents readability challenges. Specifically, the font size is inadequate or wrong, making it difficult to discern critical details and potentially obscuring important information. A revision focusing on enhanced visual legibility and appropriate font scaling would significantly improve the document's overall comprehension and professional appearance. In the correlation bubble plots the size scale could be arrange better. I can't see or read properly Figure 3, Figure 4, Figure 6, Figure 7, Figure 8 and Figure 10.
I recommend that the authors take a more time to addressing the visual presentation concerns. Specifically, they should:
- Carefully review the figure design, paying close attention to graphic resolution and legibility.
- Invest sufficient time in refining the typography, ensuring that font sizes are appropriate and easily readable across different viewing platforms.
Author Response
Thank you very much.
We have attached Figure 3, Figure 4, Figure 6, Figure 7, Figure 8, and Figure 10 in pdf and jpg format (supplementary files). Please check.
